# Churn Prediction in Gasoline Consumers under the Price Commitment Scenario

Boyang Li
*Department of Industrial*
*Engineering and Management*
*Peking University*
Beijing, China
Email: 2101112018@stu.pku.edu.cn

Yunzhe Qiu
*Department of Information Management*
*Peking University*
Beijing, China
Email: qiuyunzhe@pku.edu.cn

Lili Chen
*School of Public Health*
*Southeast University*
Nanjing, China
Email: chenlili1002@126.com

Xi Zhang*
*Department of Industrial*
*Engineering and Management*
*Peking University*
Beijing, China
Email: xi.zhang@pku.edu.cn

*Abstract*—The gasoline consumption market demonstrates significant brand loyalty, with consumers often preferring specific gas stations. To enhance customer retention and manage price fluctuation impacts, some companies have adopted price commitment strategies, assuring consumers they will not pay more than a predetermined future price, thus mitigating purchasing risks. However, predicting customer churn in the price commitment scenario is challenging due to the lack of historical churn data specific to these scenarios. To address the problem, we introduce the Enhanced Feature Adaptation Network in Price Commitments (EFANPC) model. The EFANPC model employs multi-source domain adaptation (MDA) techniques to transfer knowledge from various source domains without price commitments to the target domain with price commitments. It incorporates a newly designed loss function that considers domain distances in both price commitment and regular scenarios, effectively addressing the unsupervised customer churn prediction problem under price commitments. We develop features that reflect consumer purchasing behaviors and introduce a feature selection method combining both common and domain-specific features. This method captures the unique consumer behavior characteristics related to churn under price commitments for each source domain. Additionally, to tackle the challenges of insufficient samples and class imbalance in the source domain, we propose a method that balances class weights and utilizes samples from all source domains for each classifier's learning, enhancing predictive performance. The EFANPC model's performance is validated through a case study in North China, demonstrating its effectiveness in predicting churn and offering practical insights for gasoline companies.

*Index Terms*—Price Commitment Strategy, Customer Churn Prediction, Multi-source Domain Adaptation

## I. INTRODUCTION

In the gasoline consumption market, a substantial segment of consumers demonstrates brand loyalty, consistently choosing gas stations for refueling [1]. For gasoline retail businesses, attracting and retaining customers is essential for success.

Corresponding author: xi.zhang@pku.edu.cn

Therefore, given the impact of gasoline price fluctuations on consumer choices, and drawing inspiration from effective promotional strategies in other scenes such as ticket sales [2], goods sales [3], and order procurement [4], some companies have adopted a price commitment marketing strategy. This approach guarantees that enrolled consumers will not pay more than the committed price during a designated future period, as the actual payment price will be the lower of the committed price or the market selling price at that time. Price commitment mitigates purchasing risks for consumers [2], offering a significant assurance for their gasoline needs. Although this policy enhances consumer benefits and minimizes price volatility risks during its validity, the risk of consumer attrition persists. Therefore, identifying churn consumers under the price commitment is crucial.

In fact, the price commitment policy substantially affects consumer purchasing decisions [3]. During periods of market price increases, certain consumers, identified as price-sensitive, may opt for refueling at stations offering price commitments, thus showing a reduced propensity to switch stations. Furthermore, consumers with frequent refueling needs do not alter their consumption patterns in anticipation of price increases; instead, they tend to refuel subsequent to price reduction. Consequently, assessing the churn risk among participants of the price commitment campaign, especially in light of sales price volatility, emerges as a crucial concern that requires focused examination. Traditional churn prediction methods rely on static identity features [5] or dynamic behavior features [6]. These methods train machine learning models on historical data with churn labels and use the trained models to predict customer churn. However, predictive methodologies for consumer churn within the price commitment context are scarce, and an in-depth analysis of consumer behavior under these conditions is lacking. As this campaign is initiated for the first time, predicting consumer future behavior remains speculative,

evidenced by the absence of churn labels, as shown in Figure 1. The enterprise currently lacks direct correspondence between consumer consumption behavior and churn under the circumstances of price commitment. Therefore, leveraging existing techniques for this classification challenge proves impractical.

Nevertheless, companies possess consumer consumption data in traditional scenarios (no price commitment scenarios) from multiple gasoline stations in the local and neighboring areas. This data allows for the identification of consumer attrition based on their consumption patterns following their last transaction. Given the similarity between these two scenarios, leveraging the available data to train models and applying them to the target problem [7], which can be accomplished through the multi-source domain adaptation (MDA) technique in transfer learning (TF) [8]. The approach enables the adaptation of knowledge from multiple source domains to improve the performance on the target domain, thereby addressing the classification challenge effectively.

However, several critical issues arise in multi-source domain adaptation. Due to the absence of customer labels under price commitments, data from different sources may differ in feature space, data distribution and label. Effectively leveraging these heterogeneous data from general consumption scenarios to address the issue of customer churn prediction under price commitments is challenging. Additionally, consumption data tends to be sparse, making it difficult to identify features that positively contribute to label prediction across different domains. Furthermore, source domain data is often insufficient and imbalanced, presenting another challenge in multi-source domain adaptation.

To overcome the aforementioned challenges, we present a novel multi-source domain adaptive model Enhanced Feature Adaptation Network in Price Commitments (EFANPC) for predicting consumer churn, specifically within the price commitment. By leveraging feature transformation, we identify consumer consumption data from regions lacking price commitment initiatives as the source domain. This approach establishes a functional link between consumer behavior and churn in these regions. For each region, unique feature selection functions are developed to transpose the consumer data from the source to the target domain's feature space, which encompasses price commitment conditions. In the price commitments, considering the insufficient quantity and imbalance of classes in the source domain samples, we propose the new cross entropy function and domain distance functions. Specifically, our main contributions are summarized as follows:

1. We present the EFANPC model based on MDA, incorporating a newly designed loss function that considers domain distances in both price commitment and regular scenarios. This model effectively addresses the unsupervised customer churn prediction problem under price commitments.

2. We have independently developed features that reflect consumer purchasing behaviors and introduced a feature selection method that combines both common and domain-specific features. This method effectively captures the unique consumer behavior characteristics related to churn under price commitments for each source domain.

3. To tackle challenges of insufficient samples and class imbalance in the source domain, we propose a method that balances class weights and utilizes samples from all source domains for each classifier's learning, significantly enhancing predictive performance.

This article is structured as follows. Section II presents a review of the pertinent literature. Section III details the proposed EFANPC method. Section IV validate the performance of the proposed method through a real case study in North China. Section V gives conclusions and future work.

## II. Literature Review

In the section, we introduce the development of customer churn prediction methods. Then, we propose the effectiveness of transfer learning, including multi-source domain adaptation methods, and analyze their limitations. The specifics are detailed in the following.

### A. Churn prediction

In the contemporary, highly competitive business environment, customer churn has become a critical concern across various sectors [9]. The phenomenon of customer churn, potentially diminishing a firm's long-term profitability, highlights the essential role of accurately predicting and identifying customers at risk of churn. Customer churn prediction employs a data-driven methodology, which includes the analysis of historical data and the application of machine learning and statistical models, to anticipate which customers might discontinue their services and to implement strategies aimed at reducing churn rates. Jens et al. [10] concentrated on predicting consumer lifetime within an anonymous, location-based social network. Through extensive testing, they demonstrated that simplifying the prediction challenge to a binary decision significantly enhances the effectiveness of lifetime prediction models.

Traditionally, churn prediction analyses have predominantly focused on static customer attributes to determine their impact on the likelihood of churn. De et al. [11] explored the relationship between demographic attributes, including age, gender, income, and the propensity for churn, highlighting the significant influence of geographic location on churn behaviors. Similarly, Rajamohamed et al. [5] investigated churn prediction among credit card holders by examining the correlation between characteristics such as age, education level, marital status, and churn likelihood. Their findings reinforce the critical role that specific static features play in churn prediction.

Recent studies increasingly address the influence of consumer dynamic behavior on churn likelihood, especially under data privacy constraints that limit companies to accessing only consumers' purchase histories rather than personal identifiers like phone numbers. This highlights the importance of basing future churn predictions on consumption behavior characteristics. In churn prediction, prevalent methodologies include

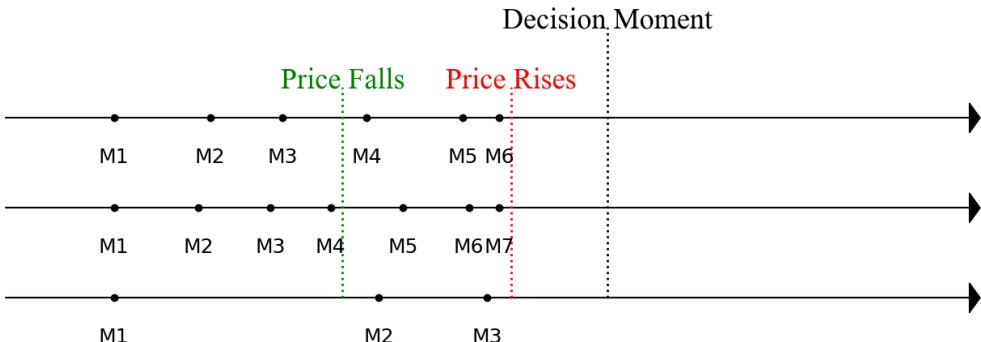

Fig. 1.  Customer consumption history and future churn window.

tree-based models, binary classification sequence models, Recurrent Neural Network (RNN)-based solutions, and survival analysis for predicting consumer return times. Wu et al. [9] introduced the Multivariate Behavior Sequence Transformer (MBST), incorporating dual attention mechanisms to analyze temporal and behavioral data independently, and integrated a tree-based classifier for enhanced churn prediction, markedly outperforming existing models. Similarly, Georg et al. [6] developed an RNN survival model, RNNSM, which surpasses traditional methods in distinguishing between returning and non-returning customers in extensive e-commerce datasets.

Nonetheless, the niche of churn prediction in the price commitment scenarios remains underexplored. Price commitments may alter consumer behaviors around market price changes, affecting the dynamic relationship between consumption behaviors and churn. Furthermore, existing research often reduces churn prediction to a binary classification challenge, overlooking the varying significance of non-churning customers. This gap necessitates a dedicated solution for predicting gasoline consumer churn in price commitment scenarios.

### B. Transfer Learning

In recent years, transfer learning has emerged as a crucial paradigm in the churn prediction, aiming to enhance the generalization capabilities and performance of models across different domains or data sources [12]. The fundamental challenge of transfer learning lies in effectively leveraging the knowledge gained in the source domain with abundant labeled data to improve learning tasks in the target domain with limited or unlabeled data. Various techniques have been proposed to address challenges related to domain shifts, taking into account differences in data distribution, feature spaces, or class labels between different domains. Transfer learning has found applications in various domains, including computer vision [13] and natural language processing [14].

Single-source domain adaptation (SDA) is a pivotal concept in transfer learning, aiming to address the knowledge transfer challenge from one domain (source domain) to another (target domain) [15]. The objective of single-source domain adaptation is to employ methodologies that enable models trained on the source domain to generalize more effectively to the target domain. This typically involves modeling the distribution differences between the source and target domains and implementing measures to mitigate these disparities [7]. One of the primary challenges is learning in the absence of target domain labels, as the target domain often lacks labeled samples [16]. Consequently, SDA methods strive to identify shared features between the source and target domains and devise strategies to counter domain-specific differences. Yang et al. [17] proposed a novel approach for deep semantic information propagation in scenarios with an unlabeled target domain and a labeled source domain, utilizing a graph attention network to achieve semantic propagation and transfer. Zhang et al. [18] introduced an optical image matching model that employs a domain adaptation (DA) method with a composite loss function to enhance the performance of unsupervised matching. However, while Single Domain Adaptation (SDA) methods often focus on a single source domain, they struggle to fully leverage multi-source information for improved adaptability. Consequently, the limitations of SDA become apparent when dealing with more complex and diverse real-world problems.

As an extension of SDA, Multi-source Domain Adaptation (MDA) deals with situations where data comes from multiple different sources or domains. Real-world data often exhibits differences in distribution, feature space, or data volume across various sources. Multi-source domain adaptation involves the integration of data from multiple source domains to enhance a model's generalization performance in a target domain [19]. This method necessitates modeling intricate relationships between diverse domains, taking into account variations among source domains, and understanding their reciprocal influence on the target domain [20]. Typically, multi-source domain adaptation methods incorporate domain alignment to mitigate distribution differences between domains, thereby improving the model's adaptability [8]. Xu et al. [21] modeled the joint distribution of observed values on different Markov networks, utilizing data from multiple source domains for label prediction. Ye et al. [22] proposed a Multi-Source Domain Adaptive Network (MSDAN) based on transfer learning, applied to battery health degradation monitoring under various operating conditions. However, traditional multi-source

domain adaptation requires a substantial amount of source data and demands a well-balanced distribution across different categories. Additionally, conventional MDA methods impose high sample quality requirements on source domain data, leading to suboptimal performance when dealing with dirty data in the source domain samples.

## III. METHODOLOGY

In this section, to enhance the accuracy of consumer churn prediction, we have meticulously crafted a churn prediction framework that is tailored to the intrinsic characteristics of the data at hand. The framework is composed of two principal components: feature extraction and the design of multi-source domain classifiers, which is shown in Figure 2. The detailed structure is delineated as follows.

### A. Notation and Problem Statement

We first present the consumption data of the $N$ gasoline customers. Among them, at every time point $t$ from 1 to cycle $T$, each customer $i$ will have a money consumption $m_{i,t}$. The information of the $i^{th}$ customer at all times constitutes the money consumption vector $\boldsymbol{m}_i = [m_{i,1}, m_{i,2}, ..., m_{i,T}]$. In fact, due to the consumption of gasoline by private cars, customers will generally wait a few days to refuel after the completion of the refuelling. Therefore, We say that transaction of consumer $i$ occurs at time $t$ when $m_{i,t} > 0$. The non-zero points $m_{i,t}$ in the matrix correspond one-to-one indicating that the refueling date of customer $i$ is time $t$, and $\boldsymbol{m}_i$ is a sparse vector.

In this problem, we need to predict the churn of customers in the future period based on their fuel consumption during the full activity cycle. Specifically, for each consumer, its daily money consumption before time $T$ are known, and its money consumption in the period from $T+1$ to $T+W$ is predicted. Mathematically speaking, when the time $T + W$ arrives, the vector $\boldsymbol{m}$ will expand on the original basis to construct a new vector $\boldsymbol{m}_i^F$ with extended consumption period, expressed as:

$$\boldsymbol{m}_i^F = [m_{i,1}, m_{i,2}, ..., m_{i,T+W}]. \tag{1}$$

In fact, according to the consumption situation in the price commitment activity cycle $T$, the churn situation and consumption level of consumers in the future window $W$ can be classified. According to the method defined in the literature [23], [24], a customer is considered to have churn when it does not continue to purchase the product for a long period of time $\Delta t$. Specifically, each customer $i$ in the dataset is labelled. Specially,

$$y^i = \begin{cases} 0, & \text{if } C(i, [T+1, T+W]) = 0, \\ 1, & \text{otherwise}. \end{cases} \tag{2}$$

In Equation 2, the label $y^i$ is set to 0 if there is no consumption during this period, and 1 otherwise. The consumption $C(i, [T+1, T+W])$ over a time interval $[T+1, T+W]$ is computed as follows:

$$C(i, [T, T+W]) = \sum_{t=T}^{T+W} m_{i,t}. \tag{3}$$

### B. Feature Extraction

According to the consumer behavior characteristics of consumers, by extracting common features and specific features between different domains, and building corresponding classifiers, the type can better adapt to the data of the target domain, improve the generalization ability, and carry out information transfer between different source domains, so as to achieve better performance on the target domain.

A shared feature extractor is a model component that is responsible for extracting shared feature representations from data from multiple source domains. This component builds a shared feature space by learning features that are common in the source domain data to capture similarities and commonalities between different source domains. Considering the sparsity of consumer consumption data, if the consumer's daily consumption amount is taken as the input, the dimension and complexity of the model will be increased, and even lead to overfitting of subsequent training. Therefore, this paper adopts the extraction method of construction features, namely:

$$\boldsymbol{x}_i' = f(\boldsymbol{x}_i), \tag{4}$$

where $\boldsymbol{x}_i = \boldsymbol{m}_i = [m_{i,1}, m_{i,2}, ..., m_{i,T}]$ represents the original consumption features of customer $i$, and $\boldsymbol{x}_i' = [x_{i,1}', x_{i,2}', ..., x_{i,n}']$ denotes the newly generated feature vector for customer $i$, with $f(\cdot)$ being the shared feature extraction function. Here, $n << T$, indicating a substantial dimensionality reduction in the original data. Specifically, referring to the feature extraction approach outlined in [25], the extracted features encompass in Table I. Upon completion of the shared feature extraction, the original consumption features from the source and target domains are transformed into smaller-dimensional feature vectors, facilitating their subsequent utilization.

Then, source-specific feature extractors refer to a feature extraction module tailored to different data sources. They are adaptable to differences among data sources and aligned with the objectives of the task, yield highly informative features that serve as valuable input for model training and prediction. Specially,

$$\boldsymbol{x}_i'^k = g_k(\boldsymbol{x}_i'), \tag{5}$$

where $g_k(\cdot)$ serves as the source-specific feature extractor for the $k^{th}$ source, incorporating the feature selection method proposed by [26]. This function effectively identifies and selects features from each source domain that are pertinent to predicting outcomes in the target domain. Subsequently, selected features are mapped into a new feature space, facilitating improved predictive performance.

### C. Multi-domain Classifier Learning

*1) Classifiers on insufficient and imbalanced data:* Following the approach proposed in [27], the process of sample filtering necessitates an evaluation of the transferability of the data from the source domain to the target domain. Initially,

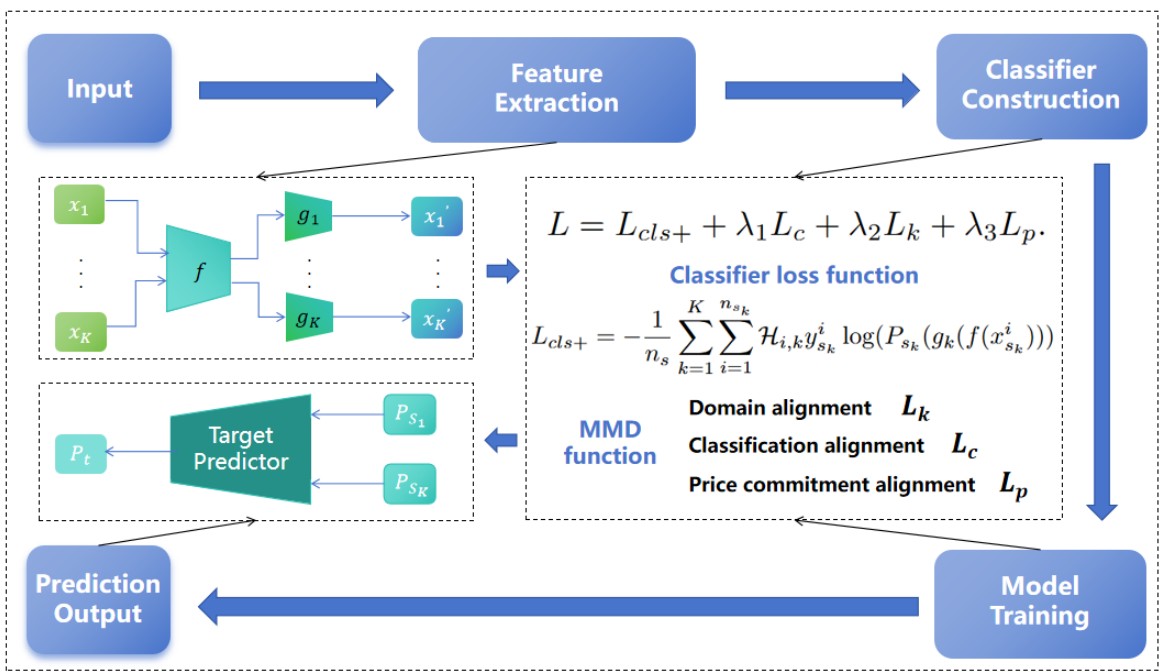

Fig. 2. Framework of EFANPC.

TABLE I
VARIABLES ABOUT THE CUSTOMERS IN THE PRICE COMMITMENT
SCENARIO.

| Variable | Index |
|---|---|
| Total Number of Purchases | $x[1]$ |
| Average Time Interval Between Purchases | $x[2]$ |
| Standard Deviation of Purchase Interval | $x[3]$ |
| Maximum Purchase Interval | $x[4]$ |
| Time Interval from Last Purchase to Present | $x[5]$ |
| Total Fuel Consumption at the Gas Station | $x[6]$ |
| Average Fuel Consumption at the Gas Station | $x[7]$ |
| Maximum Fuel Consumption at the Gas Station | $x[8]$ |
| Fuel Consumption Median at the Gas Station | $x[9]$ |
| Total Monetary Expenditure at the Gas Station | $x[10]$ |
| Average Monetary Expenditure at the Gas Station | $x[11]$ |
| Maximum Monetary Expenditure at the Gas Station | $x[12]$ |

classifiers from the source domain are individually optimized with respect to the loss function, as detailed below:

$$P_{s_k} = \arg\min L_{cls} = \arg\min_{P_{s_k}} L_{CE}(P_{s_k}(g_k(f(\boldsymbol{x}_{s_k}))), \boldsymbol{y}_{s_k}),$$
(6)

where $(\boldsymbol{x}_{s_k}, \boldsymbol{y}_{s_k}) \in \mathcal{D}_{s_k}$ and $L_{CE}(\cdot)$ is the cross-entropy loss function on labeled source data. However, when the number of single source domain samples is insufficient, it is a great choice to use all source domain samples as the training set of each specific classifier to avoid overfitting [28]. At the same time, in view of the imbalanced number of labels in the sample, in order to improve the overall prediction performance, it is significant to set different weights for labels to reduce the impact of different sample numbers [29]. Since the number of consumers with labels in each source gasoline station is

also limited and the proportion of labels is imbalanced, this paper redesigned the classification loss function to address this problem, which is specifically written as follows:

$$L_{cls+} = -\frac{1}{n_s} \sum_{k=1}^{K} \sum_{i=1}^{n_{s_k}} \mathcal{H}_{i,k} \boldsymbol{y}_{s_k}^i \log(P_{s_k}(g_k(f(\boldsymbol{x}_{s_k}^i)))), \quad (7)$$

where $K$ denotes the number of source domains, $n_{s_k}$ represents the number of samples in the $k^{th}$ source domain, and $n_s$ denotes the total number of samples across all source domains. The term $\mathcal{H}_{i,k}$ represents the class weight, which is specifically defined as follows:

$$\mathcal{H}_{i,k} = \sum_{c=1}^{C} \frac{\mathbf{1}(\boldsymbol{y}_{s_k}^i = c)}{p_{c,k}}, \quad (8)$$

where $\mathbf{1}(\cdot)$ denotes the indicator function, while $p_{c,k}$ represents the proportion of samples belonging to class $c$ in the $k^{th}$ source.

*2) MMD in the price commitment scenario:* Considering that multiple source domains are typically available, the objective is to enhance the model's performance in a target domain through adaptive techniques. The fundamental idea of Maximum Mean Discrepancy (MMD) is to maximize the discrepancy between the empirical means of the feature mappings from the two distributions $\mathcal{D}_a$ and $\mathcal{D}_b$ [30], as

expressed in Equation 9.

$$\mathcal{MMD}\left(\mathcal{D}_a, \mathcal{D}_b\right) = \sup_{h \in \mathcal{H}} \left\| \frac{1}{n_a} \sum_{i=1}^{n_a} h\left(g_k\left(f\left(\boldsymbol{x}_a^i\right)\right)\right) \right.$$
$$\left. - \frac{1}{n_b} \sum_{j=1}^{n_b} h\left(g_k\left(f\left(\boldsymbol{x}_b^j\right)\right)\right) \right\|_{\mathcal{H}}^2. \quad (9)$$

where $\mathcal{H}$ denotes a reproducing kernel Hilbert space (RKHS). Besides, $n_a$ and $n_b$ represent the number of samples from the distributions $\mathcal{D}_a$ and $\mathcal{D}_b$, respectively. The notation $\|\cdot\|_{\mathcal{H}}$ signifies the norm in the RKHS, utilized to measure the difference between the mean embeddings of the two distributions. The characteristic kernel function $h\left(\cdot\right)$ operates in $\mathcal{H}$. To ensure that the classifier trained on the source data demonstrates effective performance on the target domain, we employ a method of global alignment and category alignment between the source and target data [27], as detailed:

$$L_k = \mathcal{MMD}(\mathcal{D}_{s_k}, \mathcal{D}_t), \quad (10)$$

$$L_c = \frac{1}{C} \sum_{c=1}^{C} \mathcal{MMD}(\mathcal{D}_{s_k}^c, \mathcal{D}_t^c)$$
$$- \frac{1}{2C(C-1)} \sum_{c_1=1}^{C} \sum_{c_2 \neq c_1}^{C} \left( \mathcal{MMD}(\mathcal{D}_{s_k}^{c_1}, \mathcal{D}_{s_k}^{c_2}) \right.$$
$$\left. + \mathcal{MMD}(\mathcal{D}_t^{c_1}, \mathcal{D}_t^{c_2}) \right), \quad (11)$$

where $L_k$ is the distance of distributions on between the $k^{th}$ source domain and target domain, and $L_c$ is that on the classes. $\mathcal{D}_{s_t^c}$ and $\mathcal{D}_t^c$ are the $k^{th}$ filtered source domain and target domain in the $c^{th}$ class. Regarding the price commitment, we propose that the distribution of consumers with the same level of price sensitivity should be approximately consistent. When there are changes in gasoline prices, the same price fluctuation should have a uniform impact on consumer behavior. When selecting source domain samples, it is imperative to quantify the influence of gasoline market price fluctuations on the refueling behavior of various consumers, denoted as $Q_{s_k}^i$. This influence $Q_{s_k}^i$ is articulated as follows:

$$Q_{s_k}^i = \sum_{t=1}^{T} \mathbf{1}(t \in \mathcal{T}_c) U_{s_k}^{i,t} V_{s_k}^{i,t}, \quad (12)$$

where $\mathcal{T}_c$ represents the set of times when price changes occur compared to the previous day. $\mathbf{1}(\cdot)$ is the indicator function, and $U_{s_k}^{i,t}$ and $V_{s_k}^{i,t}$ denote the consumption interval fluctuation factor and the consumption amount factor for consumer $i$ in source domain $s_k$ at time $t$, respectively. These are specifically expressed as follows:

$$U_{s_k}^{i,t} = \mathbf{1}(\boldsymbol{m}_{s_k}^{i,t} > 0 \wedge |\Delta T_{s_k}^{i,t} - \bar{T}_{s_k}^i| > \phi), \quad (13)$$

where $M_{s_k}^{i,t}$ and $\Delta T_{s_k}^{i,t}$ respectively denote the consumption amount and the time interval between the consumption of use $i$ at time $t$ and their previous consumption, $\bar{T}_{s_k}^i$ denotes the average consumption interval for consumer $i$ within a cycle, and $\phi$ indicates the threshold for the change of consumption interval. Then,

$$V_{s_k}^{i,t} = \mathbf{1}(\sum_{t_1=t-\omega}^{t+\omega} \boldsymbol{m}_{s_k}^{i,t_1} > 0), \quad (14)$$

where $\omega$ represents the time observation radius centered around the time point when the price changes. A higher value of $Q_{s_k}^i$ indicates that the consumer's consumption behavior is more likely to be influenced by price changes, and this source sample has a greater impact on classifying consumers under price commitments. The source domain and target domain samples are divided into different price sensitivity domains, denoted as $\mathcal{D}_{s_k}'^q$ and $\mathcal{D}_t^q$, respectively. Specifically:

$$L_{pc} = \sum_{q=1}^{\max_i Q_t^i} \mathcal{MMD}(\mathcal{D}_{s_k}^q, \mathcal{D}_t^q). \quad (15)$$

By incorporating all the aforementioned elements, we propose an enhanced loss function, as detailed below:

$$L = L_{cls+} + \lambda_1 L_c + \lambda_2 L_k + \lambda_3 L_p. \quad (16)$$

*3) Domain Discrimination and Prediction:* In order to distinguish different domains, we construct the domain discriminator $F_d$ to determine which specific source domain each sample belongs to. This is achieved by minimizing the domain discrimination loss function $L_{\mathrm{dd}}$, which is defined as follows:

$$L_{\mathrm{dd}} = \sum_{k=1}^{K} L_{CE}\left(F_d\left(f\left(\boldsymbol{x}_{s_k}\right)\right), k\right), \boldsymbol{x}_{s_k} \in \{\mathcal{D}_{s_k}\}_{k=1}^{K}. \quad (17)$$

The trained domain discriminator $F_d$ can be used to assess the similarity between specific target domain samples and different source domains. The probability list of the target domain samples is defined as follows:

$$F_d\left(f\left(x_t\right)\right) = \left[p_{t_d}^1, p_{t_d}^2, \ldots, p_{t_d}^K\right]. \quad (18)$$

Here, $p_{t_d}^k$ denotes the probability that the sample belongs to category $k$. In the subsequent prediction process, the classification results for the target domain are weighted and aggregated based on the probabilities of target domain samples belonging to each source domain. Specifically, the classification predictor for the target domain $P_t$ is weighted and summed as follows:

$$P_t = \sum_{k=1}^{K} p_{t_d}^k P_{s_k}. \quad (19)$$

After the model is established, we need to train the model parameters, as detailed in Algorithm 1. Subsequently, the trained classifier is used to obtain the prediction results for the corresponding target domain samples, as shown in Formula 19.

**Algorithm 1:** The process of task domain prediction

---

**Input:** Source domains $\{\mathcal{D}_{s_k}\}_{k=1}^{K}$, target domain $\mathcal{D}_t$.
Pre-trained feature extraction networks $f, g_k$
and source predictor $P_{s_k}$;
**Output:** Trained $g_k$ and $P_{s_k}$;

1 Learn domain discriminator as in 17;
2 Calculate the weight vector of the target sample being
  an insider of source domains as in 18;
3 **for** $i = 1, i < Max\_iter, i++$ **do**
4     Refine source and target data by minimizing the
      function in 10 and 11;
5     Train the source domain model to the target
      domain using price sensitivity under the price
      commitment in 15;
6     Update $g_k$ and $P_{s_k}$ according to 16;
7 **end**

---

## IV. EMPIRICAL ANALYSIS

### A. Data Description

In this study, we utilize a dataset derived from consumer transactions at four gas stations in northern China. Three of the gas stations represent the source domain data, depicting traditional consumption patterns, while one gas station represents consumption under the price commitment scenario. The labeled dataset from these three gas stations under traditional conditions includes gasoline sales records from January 1, 2021, to August 1, 2022. The unlabeled data from the gas station under the price commitment scheme covers the period from August 12, 2022, to December 31, 2022. The datasets consist of several features, including customer ID (phone number), transaction timestamp, payment amount per transaction, and the quantity of fuel purchased per transaction. The source domain datasets comprise 62,062, 266,041, and 194,028 records from 4,954, 12,906, and 8,967 consumers, respectively. The target domain dataset includes 4,883 records from 1,338 consumers who participated in the price commitment scenario.

### B. Experimental Setting

In our experiments, we set the time observation radius $\omega$ to 5 days, the threshold of interval change $\phi$ to 5 days and the future consumption window length W to 90. Following the methodologies outlined in [27], [31], we defined $\lambda_1 = \lambda_2 = \lambda_3 = \frac{2}{1+e^{-10\rho}} - 1$, where $\rho$ is a linearly varying parameter from 0 to 1. Additionally, we employed a batch size of 128, conducted $Max\_iter = 100$ iterations, utilized a learning rate schedule with an initial value of 0.01, applied a momentum term of 0.9, logged training progress every 10 iterations, and incorporated an L2 regularization decay rate of $5 \times 10^{-4}$ to enhance model generalization.

### C. Visualization and Analysis

To provide deeper insights into the model's performance, we present several visualizations and analyses. Figure 3 illustrates

TABLE II
CONFUSION MATRIX SHOWING THE CLASSIFICATION RESULTS OF THE MODEL.

| | Predicted Positive (Non-churn) | Predicted Negative (Churn) |
|---|---|---|
| **Actual Positive (Non-churn)** | 248 | 90 |
| **Actual Negative (Churn)** | 107 | 893 |

the mapping of the some domain data both before and after applying the EFANPC model. This visualization helps to understand how the model transforms the data distribution in the different domains. Specifically, it shows how the EFANPC model aligns the target domain data and the source domain data, thereby improving the model's transferability.

In addition to the visualization, we provide a detailed performance evaluation using a confusion matrix, as shown in Table II. The confusion matrix provides a comprehensive breakdown of the model's classification results, highlighting the number of true non-churn cases, false non-churn cases, true churn cases, and false churn cases.

From Table II, it is evident that the model demonstrates a high true non-churn rate, which signifies its effectiveness in accurately identifying non-churn cases. This high true non-churn rate is advantageous as it ensures that most of the non-churn cases are correctly classified, minimizing the risk of falsely identifying loyal customers as churn cases. The model also exhibits a commendable ability to detect churn cases, as evidenced by the high true churn rate. This high true churn rate is beneficial because it indicates that the model effectively identifies those who are at risk of churning, thus enabling timely interventions.

### D. Results Analysis

In this section, we evaluate the model using Precision, Recall, and F1-Score as our metrics. Precision measures the accuracy of the model's positive predictions, Recall assesses the model's ability to identify all positive instances, and the F1-Score balances Precision and Recall, providing a comprehensive evaluation metric. Higher values for Precision, Recall, and F1-Score indicate better model performance. The specific calculation formulas are as follows:

$$\text{Precision} = \frac{TP}{TP + FP}, \tag{20}$$

where $TP$ represents the number of true positive instances (correctly identified positive cases), and $FP$ denotes the number of false positive instances (incorrectly identified positive cases). Then,

$$\text{Recall} = \frac{TP}{TP + FN}, \tag{21}$$

where $TP$ is the number of true positive instances, and $FN$ represents the number of false negative instances (positive cases that were incorrectly identified as negative). And

$$\text{F1-Score} = 2 \times \frac{\text{Precision} \times \text{Recall}}{\text{Precision} + \text{Recall}}. \tag{22}$$

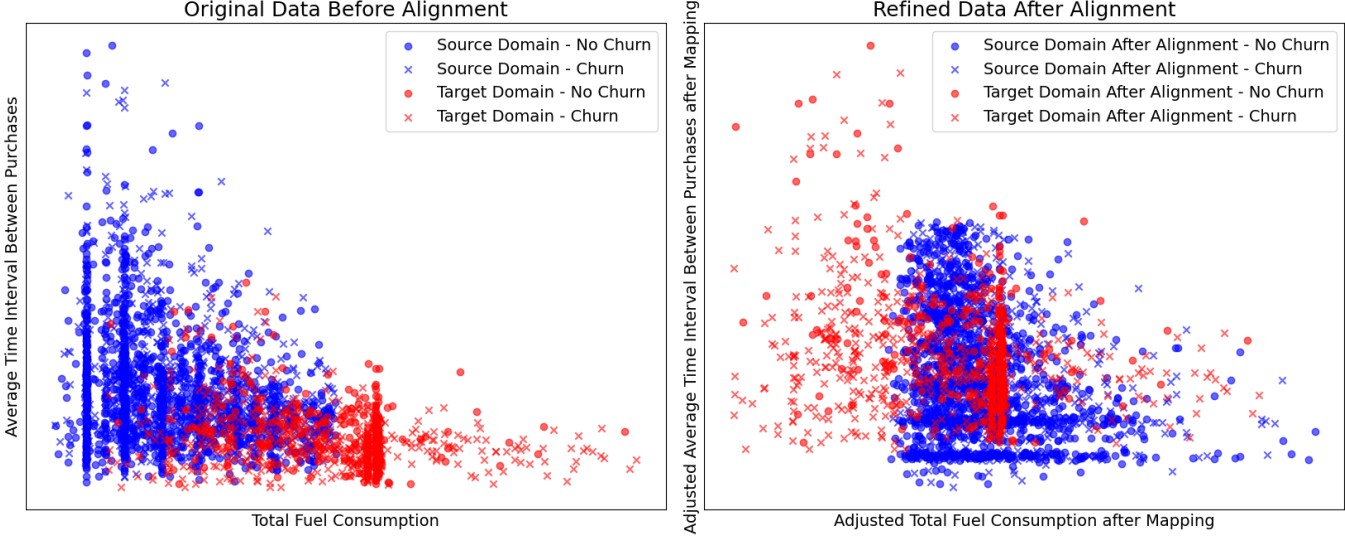

Fig. 3. Visualization of target domain data mapping before and after domain alignment.

TABLE III
COMPARISON OF MODEL PERFORMANCE IN CHURN PREDICTION BASED
ON VARIOUS EVALUATION CRITERIA.

| Model | Precision | Recall | F1-Score |
|---|---|---|---|
| MDDA [32] | 0.59 | 0.68 | 0.64 |
| MFSAN [31] | 0.62 | 0.67 | 0.64 |
| iMSDA [33] | 0.66 | 0.69 | 0.67 |
| EFANPC | **0.70** | **0.73** | **0.72** |

TABLE IV
THE ABLATION STUDY OF THE EFANPC MODEL CONDUCTED ACROSS A
SPECTRUM OF EVALUATION METRICS.

| Model | Precision | Recall | F1-score |
|---|---|---|---|
| EFANPC$^{-all}$ | 0.60 | 0.63 | 0.61 |
| EFANPC$^{-c, k, p}$ | 0.62 | 0.69 | 0.65 |
| EFANPC$^{-k, p}$ | 0.63 | 0.71 | 0.66 |
| EFANPC$^{-p}$ | 0.65 | 0.72 | 0.69 |
| EFANPC | **0.70** | **0.73** | **0.72** |

The F1-Score is the harmonic mean of Precision and Recall, providing a single metric that balances both concerns by considering their relative contributions.

In comparison with existing state-of-the-art models in the MDA domain, including MDDA [32], MFSAN [31], and iMSDA [33], our proposed model demonstrates superior classification performance across various metrics, as shown in Table III. Additionally, we conducted ablation experiments to evaluate the impact of each component on the EFANPC model's performance. Specifically, EFANPC$^{-all}$ denotes the model without the improved classification loss, optimized only using Equation 6, and excluding class alignment, domain alignment, and price commitment-sensitive consumer alignment loss functions. Subsequent ablation tests added back these loss functions incrementally. As indicated in Table IV, each module positively influences classification performance.

## V. CONCLUSION

This study presents the EFANPC model, developed to address the challenge of unsupervised customer churn prediction in the context of price commitments. The EFANPC model successfully transfers knowledge from regions devoid of price commitments, employing a novel loss function that integrates domain distances to enhance prediction accuracy. Through the design of specific consumer behavior features and the

implementation of a two-stage feature selection process, the model demonstrates significant improvements in predictive performance. A case study conducted in North China validates the effectiveness of the EFANPC model, showing superior performance compared to existing approaches. Additionally, ablation studies confirm the contribution of each model enhancement.

Future work will aim to advance the EFANPC model by focusing on several key areas. Firstly, we will investigate methods to further refine the model's adaptability to diverse and dynamic market conditions, including varying levels of data granularity and market volatility. Enhancing the model's robustness in the presence of incomplete or noisy data will also be a priority, ensuring reliable performance across different scenarios. Furthermore, we plan to explore the integration of additional features and advanced techniques, such as deep learning-based feature extraction and domain adaptation strategies, to improve the model's accuracy and generalizability. Another promising direction is the application of the EFANPC model in other domains with similar predictive challenges, such as customer retention in subscription-based services or churn prediction in the telecommunications industry. These efforts will collectively contribute to the development of a more

versatile and resilient customer churn prediction framework.

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
