# OpenReview forum: "Churn Prediction in Gasoline Consumers under the Price Commitment Scenario"
_IEEE.org/ICIST/2024/Conference — IEEE ICIST 2024 Conference Submission_

### Official Review · Reviewer_B11C · 2024-08-21
**This study presents the EFANPC model, developed to address the challenge of unsupervised customer churn prediction in the context of price commitments. Empirical analysis has demonstrated the effectiveness of the proposed plan. However, the following comments should be considered in the revision.**

**Rating:** 7
**Confidence:** 3

**Review:**

Question 1:
Please elaborate on the specifics of the newly designed loss function used in the EFANPC model? How does it effectively integrate domain distances in both price commitment and regular scenarios to enhance unsupervised customer churn prediction?
Question 2:
Please provide more details on the independent development of features reflecting consumer purchasing behaviors? How does their feature selection method effectively combine common and domain-specific features to capture unique consumer behavior characteristics related to churn under price commitments across different source domains?
Question 3:
How does the proposed method address the challenges of insufficient samples and class imbalance in the source domain? Please elaborate on how balancing class weights and utilizing samples from all source domains for each classifier’s learning enhances predictive performance in their approach?

---

### Official Review · Reviewer_Dn9F · 2024-08-21
**The paper is logically clear, the simulation results are credible, and it is recommended for publication.**

**Rating:** 8
**Confidence:** 3

**Review:**

This paper proposes the EFANPC model using multi-source domain adaptation techniques to predict customer churn in gasoline consumption markets with price commitment strategies, addressing challenges of insufficient historical data and class imbalance, and demonstrating its effectiveness through a case study. The paper is logically clear, the simulation results are credible, and it is recommended for publication. However, the following suggestions need careful consideration to further improve the quality of the paper
1.The SD-YOLO model has shown promising results in ship detection from SAR images. To further demonstrate its versatility, future research could explore extending the model to detect other maritime objects such as buoys, oil spills, or even marine life. This would not only validate the generalizability of the model but also provide valuable tools for a wider range of maritime surveillance and monitoring applications.
2.Although the SD-YOLO model achieves high accuracy on benchmark datasets, deploying it in real-world scenarios where large amounts of labeled data may not be readily available can be challenging. Incorporating transfer learning techniques by pre-training the model on a larger, more diverse dataset and then fine-tuning it on the target dataset with limited annotations could help overcome this issue. This approach would make the model more accessible and applicable in practical applications.
3.The current SD-YOLO model focuses on detecting ships in individual SAR images. However, with the availability of video SAR data, incorporating spatio-temporal information across frames could potentially further improve detection accuracy. For instance, utilizing optical flow or tracking algorithms to model the motion of ships between frames could help reduce false positives and increase robustness to occlusion or noise in individual frames. Additionally, this could enable the model to predict future ship positions, providing valuable insights for maritime traffic management and collision avoidance systems.

---

### Official Review · Reviewer_c7rL · 2024-08-22
**This paper  investigates the unsupervised customer churn prediction problem under price commitments. There are the folllwing questions need to be considered：**

**Rating:** 7
**Confidence:** 4

**Review:**

1. What are the advantages of the enhanced feature adaptation network in price commitments model in predicting customer churn? What effects will it have when combined with the MDA method?
2. What are the benefits of considering the domain distances in both price commitment and regular scenarios in the newly designed loss function? Why can the unsupervised customer churn prediction problem be solved? Please provide more details.
3. How to improve predictive performance when predicting customer churn rate in this paper? How to ensure the accuracy of the predictions in this paper?

---

### Decision · Program_Chairs · 2024-09-08

Accept (Oral)